# Connected Heterogenous Multi-Processing Architecture for Digitalization of Freight Railway Transport Applications

Markos Losada [1], Iñigo Adin [1,2,*], Alejandro Perez [1], Roberto Carlos Ramírez [1] and Jaizki Mendizabal [1,2,*]

1 CEIT-Basque Research and Technology Alliance (BRTA), Manuel Lardizabal 15, 20018 Donostia-San Sebastián, Spain; mlosada@ceit.es (M.L.); aperez@ceit.es (A.P.); rrarmirez@ceit.es (R.C.R.)
2 Electronic Engineering Department, Universidad de Navarra, Tecnun, Manuel Lardizabal 13, 20018 Donostia-San Sebastián, Spain
* Correspondence: iadin@ceit.es (I.A.); jmendizabal@ceit.es (J.M.); Tel.: +34-943212800 (I.A.)

**Abstract:** The digitalisation of freight rail is an essential improvement to create modern functions that offer a cost-effective, attractive service and improved operational opportunities to operators. These modern functions need intelligence, detection, actuation and communications. For this, generally, it is possible to process raw data in the Edge and send meaningful data over a communication link. However, the power supply is not granted in a freight wagon and so low power strategies need to be adopted. This paper presents the implementation and testing of a wireless connected heterogeneous multiprocessing architecture. From the power consumption point of view, this system has been stressed by means of a generic FFT function to evaluate the different on-board computing devices that have been decided. From the communication point of view, the LPWAN LoRa technology has been tested and validated on robustness and coverage. Thanks to the heterogeneous nature of this architecture and its configurability, it allows us to propose the most suitable computing ressources, data analysis and communication strategy in terms of efficiency and performance for the functions that this wagon on board unit needs to host and support. With this approach, operation data are reported to the centralised freight driver assistant system.

**Keywords:** railway; freight; digitalisation; multi-processors; LoRa; power assessment

## 1. Introduction

This paper aims to assess the electronic systems deployed onboard a railway freight wagon to pursue the digitalisation of all the railway transport applications. That is a real challenge on freight transportation with practical implications on communication technologies and the power consumption of devices with no power network on the wagons. That is a key pillar for the Innovation Programme 5 of the Shift2Rail initiative. Its foundations state that the cost competitiveness and reliability of freight services need to be improved considerably if the sector is to meet the ambitious objectives set in the Transport White Paper [1], in terms of developing rail freight: almost doubling the use of rail freight compared to 2005, achieving a shift of 30% of road freight over 300 km to modes such as rail or waterborne transport by 2030, and of more than 50% by 2050. Rail freight must be in a position to offer a cost-effective, attractive service to shippers that helps to shift freight away from the already-congested road network. The challenge is twofold: (a) to acquire a new service-oriented profile for rail freight services based on excellence in on-time delivery at competitive prices, interweaving its operations with other transport modes, addressing clientele needs by incorporating innovative value-added services, among others; and (b) to increase productivity by addressing current operational and system weaknesses and limitations, including interoperability issues, by finding cost-effective solutions to these problems, optimising existing infrastructure, and fostering technology transfer from other sectors into rail freight.

From that base, the Shift2Rail IP5 innovations and ambitions stand on five technology demonstrators (TD5.X), from which the first one is called 'Fleet Digitalization and Automation' [2]. This TD5.1 aims to improve strategic areas of rail freight transport by developing key technologies to enable a digital and automated rail freight system. TD5.1 includes core topics like Condition-based Maintenance (CBM), Automatic Coupling, Freight Automatic Train Operation (ATO) and Connected Driver Advisory Systems (C-DAS). Other systemic topics, e.g., automatic train preparation, are subordinate topics in these innovation fields. In addition, TD5.3, entitled "Smart Freight Wagon concepts", is concerned by these modernised functions which are compulsory towards the smart wagon concept. The rolling stocks need intelligence, sensing, actuation and communication, requiring power consumption, not granted in a freight wagon. This is why the assessment of the communication technology characteristics and power consumption depending on the computing and communication components is crucial.

It is worth mentioning here that the outcomes from this paper are not restricted to the railway sector as the digitalisation is an asset required on every sector such as industrial, energetic, transport, health, etc.

This paper presents a connected heterogeneous multiprocessing architecture which has been designed, implemented and tested for the freight railway applications explained in the following sections. Additionally, it presents conclusions after having stressed the system aiming at assessing the power consumption, performance and data transfer and coverage. The paper is structured in eight sections, starting from this introduction. Section 1 presents the more meaningful related work for the steps that are presented by the authors, then Section 2 describes the high-level freight railway case study that led to the design and analyse of the system. Section 3 details the design and implementation of the proposed system architecture and the electronic platform, followed in Section 4 by the performance on the power consumption point of view. The results are accompanied by test setup, methodology and performance evaluation. Section 5 shows the results from the tests carried out with the LoRa network in a real environment. Finally, the discussion and the conclusions are presented in Section 6.

## 2. Related Work

The freight railway digitalisation presents challenging scenarios due to its wide geographical distribution, harsh environmental conditions and strict energy awareness. Effective digitisation of a freight train requires sensor networks that provide online information on the status of the train in a reliable, safe and efficient way. The deployment of wireless onboard sensor networks entails efficiently managing the vast amount of data they generate and operating in very restricted energy availability.

The development of technologies related to the Internet of Things (IoT) paradigm, especially Low-Power Wide-Area Networks (LPWANs) protocols, has enabled a whole set of monitoring applications. Commercially available LPWAN technologies, like LoRA, are promising for implementing Wireless Sensor Networks (WSN) for remote rail assets monitoring applications [3]. However, the low-power consumption characteristic of LPWAN is partly due to their communication protocol with short messages and the fact that these do not need to be transmitted constantly. Therefore, LoRA technology is not viable in applications where monitoring with a transfer of large amounts of information in real-time is required. On the other hand, intelligent cloud-based LPWAN monitoring systems present important limitations such as unpredicted latency in safety-critical and performance-sensitive applications [4].

Edge computing is a paradigm that addresses the efficient handling of data generated by sensor networks. This vision change implies reducing network load and improving performance by bringing computation and storage near the data source. Under the Edge computing paradigm, specialised embedded systems are used to perform some level of data processing. In this way, the system's intelligence would not depend exclusively on cloud processing or communications. Latency and power consumption are reduced by



spacing out the frequency and load of transmissions, and security and privacy preservation are improved.

In recent years, different approaches have been taken to the problem of integrating railway monitoring systems efficiently. Various IoT and edge computing implementation projects have been proposed for freight rail applications. Jo et al. introduce a cost-effective IoT solution consisting of device platform, gateway, IoT network and platform server for smart railway infrastructure. The study evaluates and demonstrates the applicability of IoT-based maintenance through a case study that includes a proof of concept and field tests. For delivering IoT data, the authors propose a network architecture and evaluate the power consumption and coverage aspects of LTE and LoRa technologies [5]. Liu et al. have proposed an improved LoRa system composed by Edge Computing at the node connected to a LoRa gateway, which decreases the message response time of the system by moving some services to the gateway [6]. Väänänen and Hämäläinen present a survey that goes through many aspects to consider in edge and fog devices to minimise energy consumption and thus lengthen the device and the network lifetime [7]. Bernal et al. present a recent example of applying an IoT monitoring system with Edge Computing in a railway environment [8]. This system continuously monitors train running status parameters using a wireless rail monitoring system. A way-side wireless sensing system collects vibrations of rails under the influence of the wheels of passing trains. Edge Computing is used to analyse the data and calculate the state parameters of the vehicle with an integrated microprocessor. Then, the final analysis results are uploaded to a cloud server via Narrow Band Internet-of-Things (NB-IoT).

Despite the intense interest in energy efficiency in cloud data centres, Edge Computing has been largely unexplored due to the complicated interactions between edge devices, edge servers and cloud data centres and due to the absence of energy supply in remote wagons such as the vast majority of freight transport application. Recently, Jong et al. presented a review of the edge devices, edge servers, and cloud data centres [9]. The article outlines the state-of-art research on energy-aware Edge Computing and discusses related research challenges and directions, including operating systems, middleware, applications and computation offloading. As the authors point out, Edge Computing demands low-power, self-reprogramming, self-reconfigurable and rewritable devices in a heterogeneous environment. Thus, re-programmability and reconfigurability allow adapting energy consumption to different hardware and software platforms. They explore this hardware adaptability proposing and testing a device equipped with configuration buttons for different running OS, reconfigurable for different Edge Computing scenarios. Chernov et al. present a possible implementation of a smart object concept with reconfigurable hardware. An implementation of a rough-sets-based controller for data pre-processing for rail infrastructure management is included in the reference [10].

The energy and power efficiency of underlying hardware require to be explored and investigated before deploying Edge Computing infrastructure. Edge Computing platforms need to be built with energy awareness to meet application demands while minimising energy consumption and cost. A trade-off need to be taken between the duration of the event to be intercommunicated and the intensity of the use. Morabito performs a detailed evaluation of the performance of several Single-Board Computer platforms commonly used in IoT systems [11]. The aim was to conduct an extensive performance evaluation to assess the feasibility of efficiently deploying virtualised instances on single-board computers. Other authors present a novel time-energy-cost analysis of wimpy Edge Computing compared to traditional brawny cloud computing [12]. This analysis uses heterogeneous systems and selects six representative MapReduce applications applicable to IoT Edge Computing. They point out that using wimpy systems as Edge Computing devices saves costs compared to traditional cloud computing.

These latest mentioned references to edge computing highlight the convenience of reconfiguring hardware systems to achieve greater energy efficiency. However, the conve-

nience of reconfigurable hardware, based on heterogeneous multiprocessing architecture with microcontrollers, has not been explored.

## 3. System Applicability

With the Edge Computing and the platform processing and communication configurability as the main guides and considering the need to keep on progressing on the modernisation of the freight transport sector, this paper introduces the digitalisation trends of the freight railways. It presents the analysis performed with the implementation of the digitalisation and automation functions into a multi processing, and the conclusions obtained from the communication point of view when given functions are deployed into this platforms that interconnects digital cyber-physical systems on each wagon and the locomotive.

From a taxonomy point of view, the functions needed for the digitalisation of the freight railways could be categorised into several concepts and objectives, as presented in the following subsection. On the practical side, the Wagon On Board Units (WOBU) are installed on the wagons and they are all interconnected for the variety of functions deployed on each wagon and on the complete train. In addition, auxiliary electronic systems are deployed in the wagon to actuate and sense on the elements, as illustrated in Figure 1, where WOBU stands for Wagon On Board Unit and BOBU stands for Bogie On Board Unit. The functions executed with the system on board that are listed in the next subsection have different level of needs for processing, communications and latency, as will be explained below.

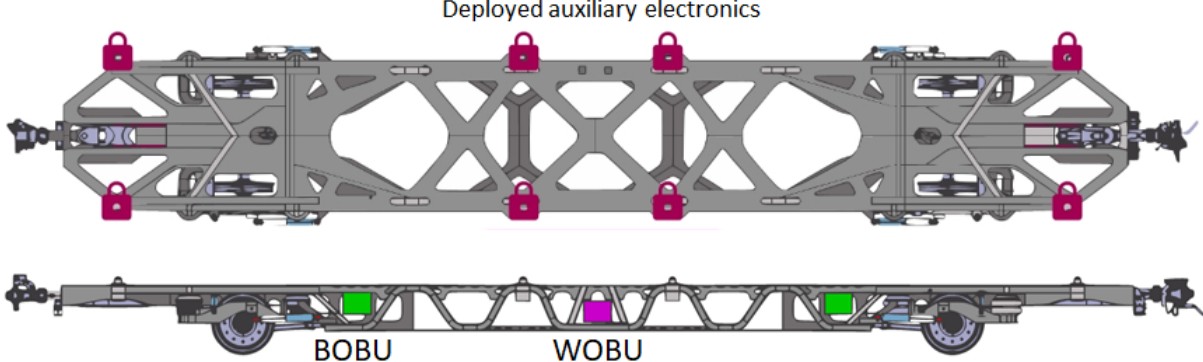

**Figure 1.** Telematic subsystems deployed on a freight wagon, by example of the Extended Market Wagon developed in the project FR8RAIL-4.

*Case Study: FR8RAIL*

We can distinguish safety and non-safety functions from the list shown in Table 1. From the communication and processing point of view, the safety ones have higher frequency monitoring and reporting data than the others, for which the control and access to the data is more reactive. The time window for these functions is established by the procedures of the freight railway sector. However, in some cases, automation is still a duty of the standardisation committees as they are not even proposed. As part of the step forward proposed by this paper, a connected heterogeneous multiprocessing architecture could fulfil the technical requirements for the functions mentioned above. This platform could be helpful to test different architecture configurations to assess the real needs of the functions and conclude how they should be designed primarily in terms of its scope, communications characteristics, processing power and technology. The heterogeneous multiprocessing platform is based on several processors, with and without an operating system and with and without memory allocation, as will be explained in the next section.

The following are the most representative functions for digitalisation and automation identified by the main stakeholders around the freight railway transport, as proposed by the working teams in IP5 Shift2Rail projects. In this list, the intensity of the processing power required is nuanced from (+) as low to (++) as medium and to (+++) as intensive.

**Table 1.** Functions for digitalisation and automation around the freight railway transport.

| Main Functions | Sub-Functions | Processing Power |
|---|---|---|
| **Train composition** | -Command: automatic or manual from driver desk<br>-Number of wagons<br>-Order of wagons into the train | (+)<br>(+)<br>(+) |
| **Train integrity** | -Status<br>-Alarm on train integrity breach | (++)<br>(+) |
| **DAC—Digital Automatic Coupler** | -Status<br>-Command: uncoupling | (++)<br>(+) |
| **Wagon Monitoring System** | -WMS sensors info | (+++) |
| **Automatic Brake Test** | -Status<br>-Command: Hand brake | (+++)<br>(+) |
| **Pressure level control** | -Status<br>-Command: Increase/decrease pressure level | (+)<br>(+) |
| **Twist-locks** | -Status<br>-Command: lock-unlock | (+)<br>(+) |
| **Cargo Monitoring System** | -CMS sensors info | (+) |

Two of the most representative and stringent cases from this table are analysed. Wagon Monitoring System and Cargo Monitoring System have been proposed and selected for the rationale and analysis of this paper. They balance a low latency data collection, real-time data analysis and communication between subsystems, which, in sum, gather the canonical utilities on digitalisation. The inputs to evaluate the information provided by the sensors deployed could be faced from several technologies. Still, it is coherent to consider that a Fast Fourier Transform (FFT) function is plausible from the processing point of view, for the accelerometers. For this case of Wagon Monitoring System, it will be designed and demonstrated for the multiple processing units included in the platform presented in this publication. Moreover, proper communication technology should accompany this data, for the messages and commands coming from every wagon to reach the locomotive unit, where the driver-assisted management system is connected.

## 4. Digitalisation of the Freight Railways

A centralised wireless architecture is proposed for the freight train digital composition and control of the functions, as seen in Figure 2. The subsections below present the HW, FW and SW definition of the Wagon On-Board Unit system deployed in each wagon, which is a connected multiprocessing platform. In the next figure, LOBU stands for Locomotive On-Board Unit, FTSMS stands for Freight Train Status Monitoring System, and FC-DAS stands for Freight Convoy-Driver Assistance System.

In the proposed computing architecture for the WOBU elements, a hardware (HW) and firmware (FW) implementation are suggested, as seen in the following subsections.

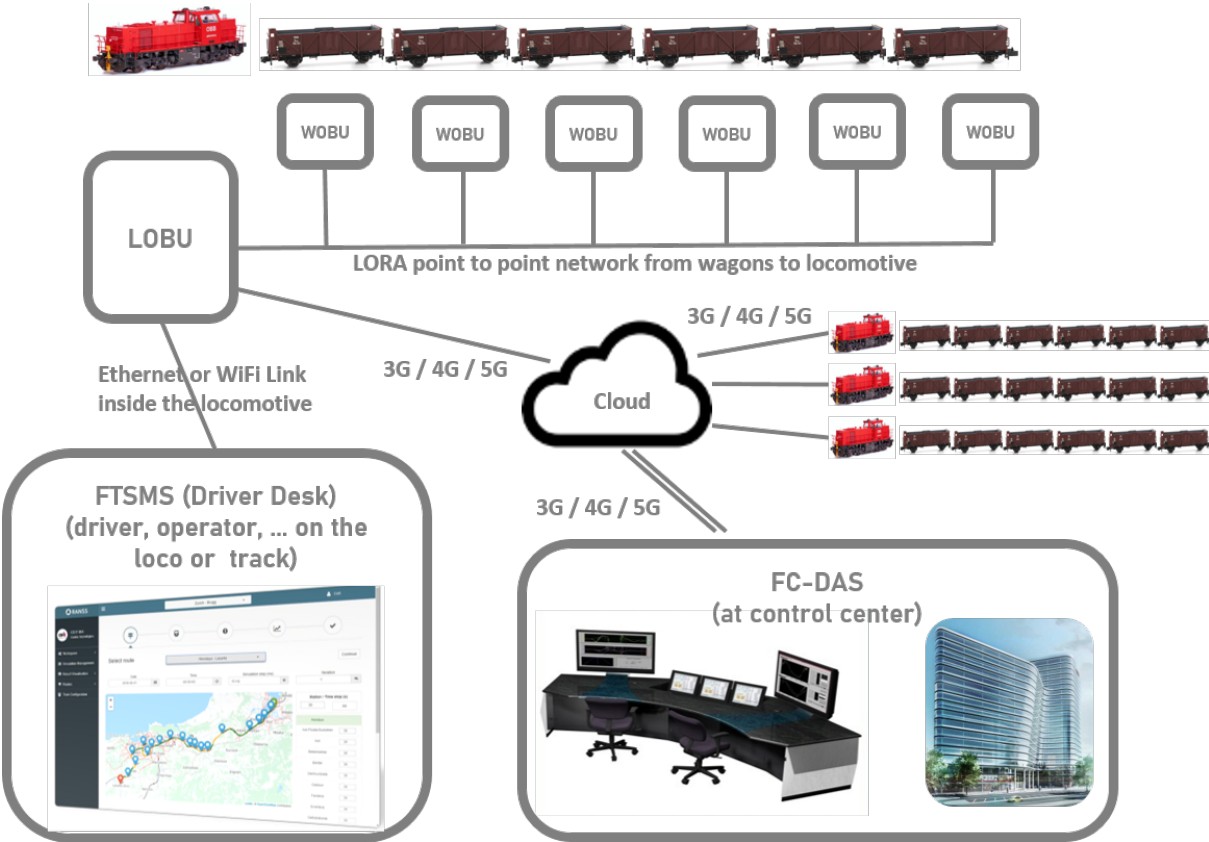

**Figure 2.** Full digital system integration of the freight railway sector.

### 4.1. Proposed HW Implementation

Figure 3 shows the low-level block of the HW of the proposed connected computing architecture. This architecture consists of three controllers, which are a mainstream microcontroller (ST, blue block), a crossover controller (iMXRT, green block) and a system on module (SOM) (iMX8-based, orange block), a series of devices for sensorisation or communications (purple blocks) and the power supply system (yellow blocks).

A custom card is designed which has a SODIMM type connector to connect a VAR-SOM-MX8M-MINI [13] and also another to connect an iMXRT1010 evaluation board (EVB) [14]. A certified railway connector for the railway field has been included to wire a can bus and RS485 bus, which complements the Ethernet and USB interfaces. Both types are expandable on the enclosure defined and implemented. Figure 4 shows a photo of the designed card. Mechanical dimensions are 150 mm × 95 mm × 45 mm.

- **Controllers:** The main controller is the STM32F105RC [15] microcontroller, which is responsible for enabling the power supplies of the other two controllers. This controller is the cornerstone of the power management to centralise the assessment of the energy consumption on the various configurations of this heterogeneous platform. This ST microcontroller is connected to all sensors and communication devices. This controller incorporates an ARM Cortex-M3 32-bit RISC core operating at 72 MHz frequency, 256 Kbytes of Flash memory and 64 Kbytes of RAM. It is programmed through a JTAG interface, and a micro USB type B connector provides a USB CDC class interface.

  In the iMXRT-EVB lies at his heart the iMXRT1010 [16] crossover controller. This controller incorporates an ARM Cortex-M7 core operating at 500 MHz frequency and 16 Kbytes of SRAM. The 16 Mbytes flash memory is integrated into the evaluation board and accessed by the controller through a QSPI interface. A micro USB type B connector provides an UART interface for flash programming purposes and a virtual com port for data output.

The VAR-SOM-MX8M-MINI is based on the i.MX 8M Mini [17] 1.8 GHz Quad-core ARM Cortex-A53 processor. The SOM includes 16 Gbytes of eMMC Flash memory and 2 GB DDR4. A Debian distribution Linux operating system (SO) is embedded in the iMX8-SOM. The SO runs from the external SD, and a micro USB connector provides access to the virtual com port of the SO through an UART interface to interact with it. These three controllers can manage the power supply that feeds the sensors and communication devices.

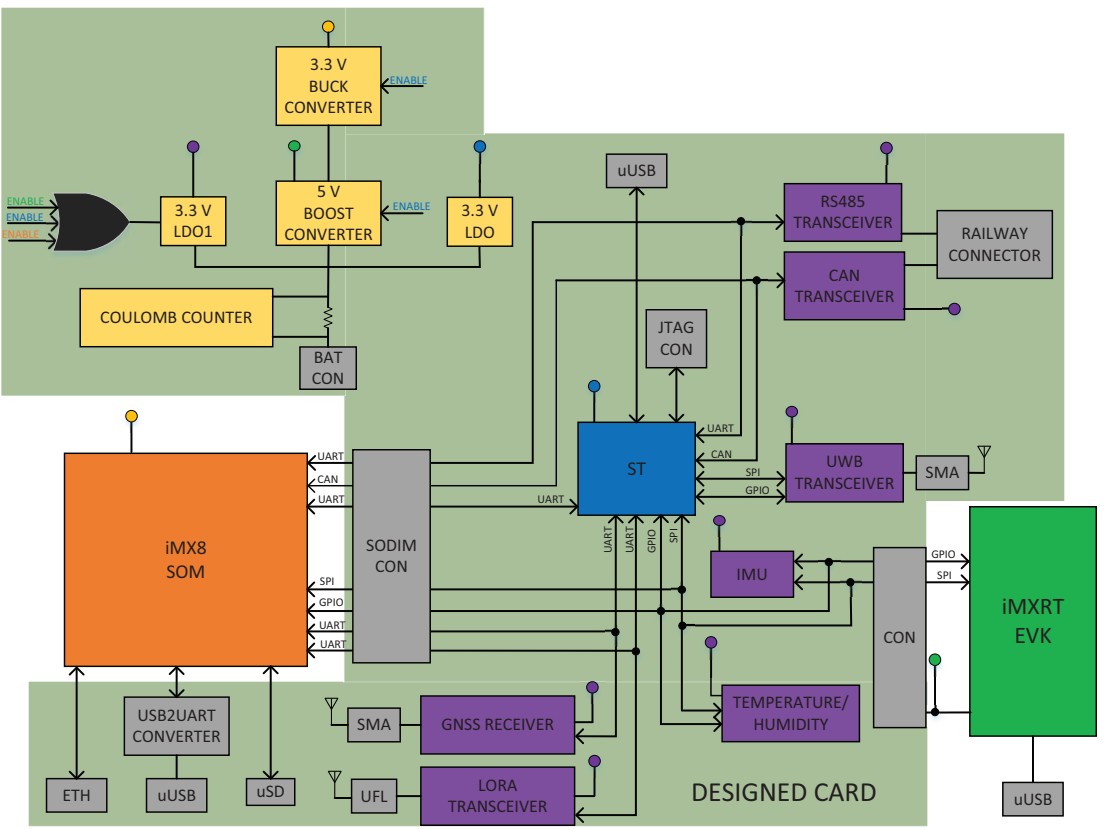

**Figure 3.** Block diagram of the proposed HW.

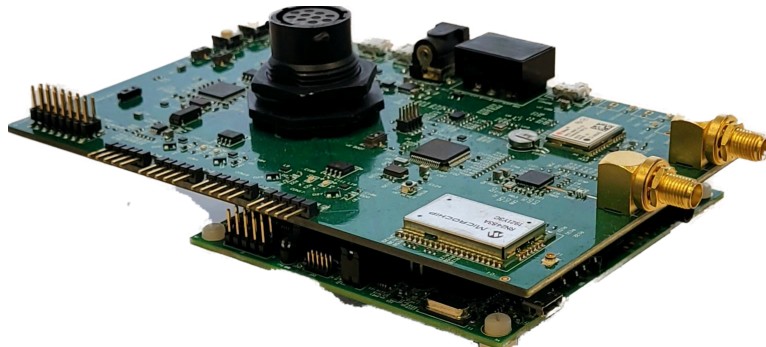

**Figure 4.** Photograph of the designed HW.

- **Communications:** A fully certified 868 MHz module based on wireless LoRa technology has been included through the transceiver module RN2483 [18]. The RN2483 module's embedded LoRaWAN protocol enables seamless connectivity to any LoRaWAN compliant network infrastructure. A UFL connector is deployed to attach an 868 MHz antenna by an SMA connector on the enclosure. The processors can interact with the LoRa module using an UART interface.

The IR-UWB technology based on the IEEE802.15.4-2011 standard has been incorporated into the designed card through the DW1000 [19] compliant low-power and low-cost transceiver from Decawave. An SMA connector has been deployed to use an omnidirectional UWB antenna. As mentioned before, the main controller interacts with the IR-UWB transceiver using an SPI interface. In addition to wireless connectivity, the iMX8 has been equipped with an Ethernet port.

- **Sensors:** The LSM6DSL [20] MEMS sensor IC is included in the designed card. It features a 3D digital accelerometer and a 3D digital gyroscope. The LSM6DSL has a full-scale acceleration range of up to ±16 g and an angular rate range of up to ±2000 dps. Moreover, a 4 kbyte FIFO memory is available for dynamic data batching. It is connected to the controllers through an SPI interface.

  The BME280 [21] humidity, barometric pressure and ambient temperature sensor is also integrated. It features a low power consumption in sleep mode and an SPI interface.

  The NEO-M8T [22] module receiver has been included in the designed card to support BeiDou, GLONASS and Galileo constellations. An active antenna that is connected to an SMA connector is designed. The ST and iMX8-SOM can interact with the GNSS module using an UART interface. As a note, the UWB technology could also be used as a sensor for the diverse functions foreseen on the freight railways' digitalisation.

- **Power supply:** The main regulator is a Low Drop Out (LDO) regulator that feeds the ST microcontroller at 3.3 V. This regulator is always ON and the ST microcontroller. Another LDO (LDO1) regulator provides sensors and communication devices at 3.3 V. Any microcontroller can enable this regulator. A boost converter switching regulator increases the voltage up to 5 V to power the iMXRT-EVK. This regulator output is connected to a buck switching regulator whose output is set at 3.3 V to power the iMX8-SOM. Note that the ST microcontroller controls both switching regulators.

  The main power source for the current development is a battery with a nominal voltage of 3.7 V and with a capacity of 10,000 mAh. A coulomb counter is included in the card to monitor the battery status. The controllers can interact with the Coulomb counter using an I2C interface.

### 4.2. Proposed Computing Architecture FW implementation

Given the centralised locomotive architecture of the digitalised freight train composition, LoRa is selected for the communication among WOBU units and LOBU with no hops to avoid limitations due to malfunctioning. To test this LoRa communication of the designed system, a firmware was developed in the ST microcontroller using only the LoRa radio layer at the 868 MHz frequency band. The device installed on the locomotive—LOBU—is used as a receiving node (Master node). In contrast, five WOBU devices are configured to obtain the data from the sensors, process the data, and send it to the Master node, where the data obtained would be stored and communicated to the FTSMS and the cloud infrastructure.

- **Master Node:** Figure 5 shows Master node workflow that is limited to configuring the LoRa chip in reception mode. It processes the data and stores the information when it receives a message. Once the data has been correctly saved, the reception mode is configured again in the LoRa chip, and it waits to receive the following message to repeat the process.

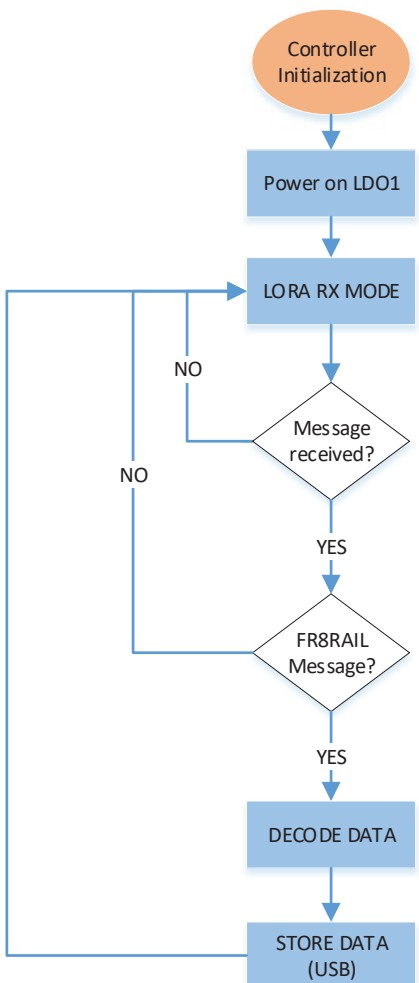

**Figure 5.** Flow chart of the Master node implementation.

- **Sensor Node:** Figure 6 shows the workflow for the sensor nodes. If it is the first boot, the node performs a clock synchronisation routine. To do this, it activates the GNSS receiver. It extracts the time values obtained from the Navigation NAV-PVT (Position, Velocity and Timing) message provided by the sensor to write them in the internal clock of the ST microcontroller and the backup registers. If it is not the first startup of the sensor node, the reading of the GNSS sensor is disabled, and the synchronisation of the internal clock of the ST is directly performed by reading the values stored in these backup registers. Once the sensor nodes are synchronised, the data obtained from the sensors, such as the temperature, humidity sensor and the accelerometer are read, this data is converted to hexadecimal format and packed. Subsequently, the LoRa chip is configured in transmitter mode, and the packet is sent. Finally, the clock values are read to calculate the time that the node must remain in sleep mode until the next cycle of measurements begins.

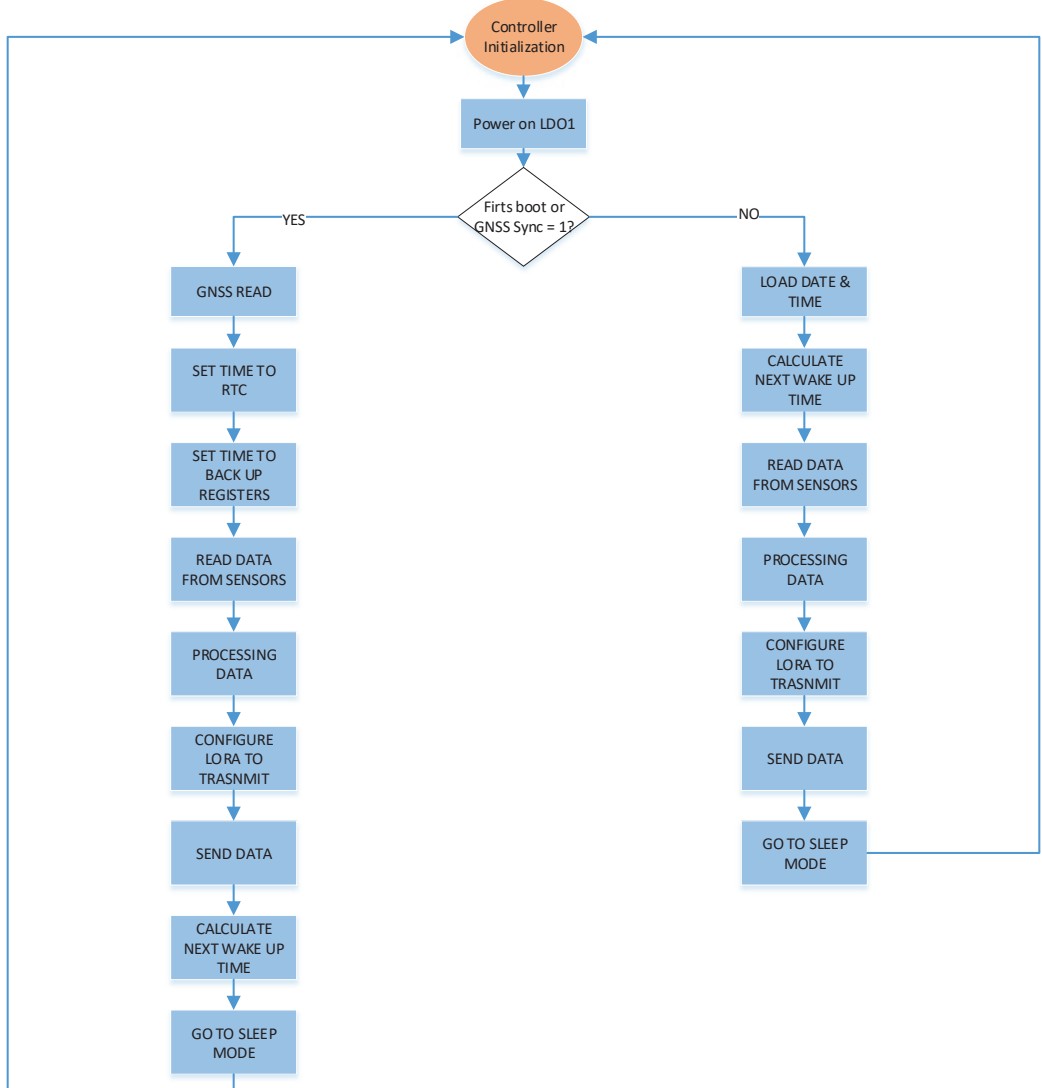

**Figure 6.** Flow chart of the Sensor node implementation.

- **Processing**

  Within the framework of the tasks or processes that the digital and automated rail freight system might have to execute, to allow us to analyse the performance and power consumption of the different controllers, a Fast Fourier Transform (FFT) of the acceleration data for the X, Y and Z axes is implemented. Figure 7 shows the flow chart of this implementation, agnostic to the processor until the step that will execute it. Note that the objective is not to analyse the performance of the implemented FFT but to use that implementation in the three controllers to evaluate their performance and consumption; hence, a specific description of how the FFT has been implemented is not carried out.

  At startup, the controller initialisation is carried out with the parameters preconfigured. The controller enables the LDO1 that feeds sensors and communication devices. Afterwards, the IMU is configured according to the user-specified parameters while wireless communication devices are set in shutdown mode. Finally, the timer configuration is done. Once at this point, the initial configuration has finished, and an infinite loop starts.

  The loop begins when the controller timer and the IMU's FIFO starts. The timer is set with the time window ($T_w$) to be measured according to the user-specified parameters,

and thus the timer flag interrupt will be triggered when this time is elapsed. The FIFO interruption will be triggered when it is full of data acquired by the IMU. If none of these interrupts is triggered, the controller will be in a low power state (IDLE mode for RT, STOP mode for ST and nothing for iMX8, which does not have low power mode) while the IMU will be capturing and saving samples in the FIFO. If the FIFO interrupt is triggered, the entire FIFO memory block will be immediately read. Once it is read from the FIFO, it is cleared. It will be stored in the controller's RAM, and then the controller will return to the low-power state. If the timer interrupt flag is set, the data found in the FIFO will be read up to that moment. At this point in the process, the acceleration data for all three axes during that time window ($T_w$) is in the controller's RAM. In the case that the off-time ($T_{OFF}$) is 0, the controller will process the FFTs of the three axes, and it will return to the low power mode. The IMU does not stop in this case, so the FIFO continues filling with new data while the controller processes the FFTs. In the case that the $T_{OFF}$ is greater than 0, the controller will disable the LDO1 and then it will process the FFTs. Then an alarm will be configured to exit from ultra-low power mode (LPIDLE mode for RT, LPSTOP mode for ST and nothing for iMX8, which does not have ultra-low power mode) when the $T_{OFF}$ expires. When this time passes, the LDO1 will be re-enabled, the IMU will be reconfigured, and the loop will be relaunched.

As a note, $T_{OFF}$ is the time in which the node is in sleep mode when there is room for determining a duty cycle on the operation of the function.

The FIFO memory has a size of 4096 bytes. Each sample of the IMU consists of 2 bytes either from the accelerometer or from the gyroscope. This means that the FIFO will store up to 341 sets of accelerations $X,Y,Z$ and angular velocities $\theta_x$, $\theta_y$, $\theta_z$, i.e., $4096 \div 2 \div 6 = 341$ set of samples. Hence, the time it takes to fill the FIFO is $\frac{341}{F_s}$. The time window ($T_w$) is set according to the user-specified number of FFT points (nFFT) and the sampling frequency ($F_s$), that is $T_w = \frac{nFFT}{F_s}$. Table 2 shows the time it takes the IMU to fill the FIFO and the duration of the time window ($T_w$) as a function of $F_s$ and the nFFTs.

**Table 2.** FIFO FULL time and time window ($T_w$).

| $F_s$ (Hz) | FIFO-FILL (s) | $T_w$ (s) | | | |
|---|---|---|---|---|---|
| | | (nFFT = 128) | (nFFT = 256) | (nFFT = 512) | (nFFT = 1024) |
| 104 | 3.27 | 1.23 | 2.46 | 4.92 | 9.84 |
| 208 | 1.63 | 0.615 | 1.23 | 2.46 | 4.92 |
| 416 | 0.815 | 0.307 | 0.615 | 1.23 | 2.46 |
| 833 | 0.407 | 0.153 | 0.307 | 0.615 | 1.23 |
| 1660 | 0.203 | 0.076 | 0.153 | 0.307 | 0.615 |
| 3330 | 0.101 | 0.038 | 0.076 | 0.153 | 0.307 |
| 6660 | 0.050 | 0.019 | 0.038 | 0.076 | 0.153 |

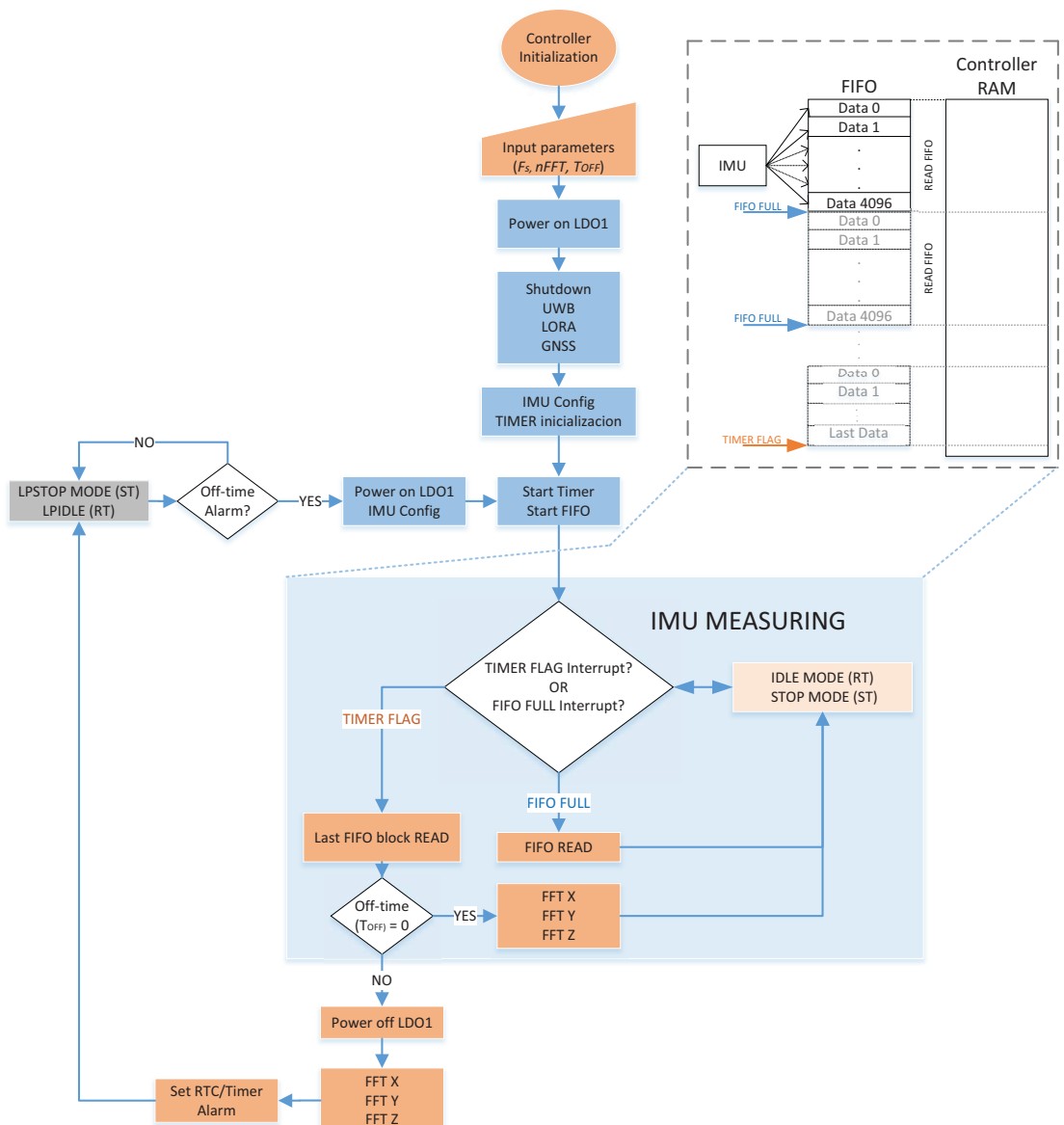

**Figure 7.** Flow chart of the proposed processing implementation.

## 5. Performance and Power Consumption Results

As previously stated, the proposed computing architecture allows executing the different tasks or processes required by the digital and automated rail freight system in the most suitable controller depending on the performance and consumption to be achieved. In this case, the results presented below are intended to show the capabilities in terms of consumption and performance for the processing explained in Section 4.2. Various parameters will be modified in this study to identify which controller is the most appropriate for this processing in terms of consumption or performance.

### 5.1. Setup

This study aims to measure the power consumption generated by the proposed computing architecture and its performance. To do that, we have implemented the setup shown in Figure 8. The same setup configuration has been implemented for all the cases to measure power consumption and performance.

The Otii Arc [23], power analyser and power supply, has been used to set a 3.7 V power supply and also monitor the current consumption. This tool has been used as a

profiler to measure currents with a high input range. It can measure magnitudes from 5 amps down to tens of nanoamps with a sample rate up to 4 ksps. A desktop application has been connected to the Otti Arc USB port to record and display the measured currents in real-time. Moreover, we have synchronised the debug logs of the proposed computing architecture with the Otii Arc shown current and voltage graphs, making it easy to correlate power consumption to the application state. Performance measurements are displayed on the PC console through the corresponding interface of each controller.

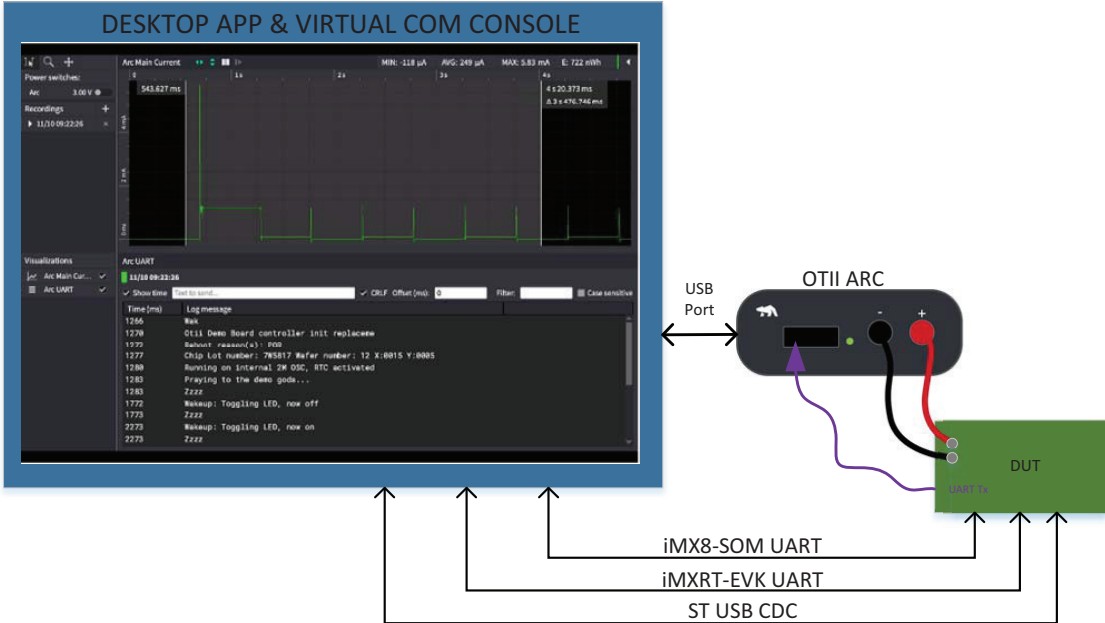

**Figure 8.** Implemented setup for the power consumption measurements.

*5.2. Methodology*

The FW operation for the proposed architecture is described in Section 4.2. The duty cycle is the relationship between the run time ($T_{ON}$), the time in which the nodes are working, and the cycle time ($T_C$), which is the total time of one cycle. As a reminder, $T_{OFF}$ is the time in which the node is in sleep mode. This mode is the lowest possible power mode so that the RAM is not lost and the controller can return to the run mode. The obtained average power consumption ($\overline{P}$) is shown as a function of off-time ($T_{OFF}$) by the following Equation (1), such as:

$$\overline{P} = \frac{(T_{ON} \times \overline{P_{ON}}) + (T_{OFF} \times \overline{P_{OFF}})}{T_C} \tag{1}$$

where $T_{ON}$ and $\overline{P_{ON}}$ are the time duration and average power consumption of the run mode, respectively. $T_{OFF}$ and $\overline{P_{OFF}}$ are the time duration and average power consumption of the off-time, respectively.

Figure 9 presents the setup used for the measurement of the average power consumption ($\overline{P}$) of the proposed system. It is sent to sleep mode when $T_{OFF} is different to 0$. The average power consumption of the off-time ($\overline{P_{OFF}}$) is also measured.

A consumption profile of the proposed architecture is illustrated in Figure 9 for the case of $T_{OFF} > 0$ and in Figure 10 for the case of $T_{OFF} = 0$, which are related to the flow chart shown in Figure 7. These illustrations are an example of three times full FIFO readings. Note that we have used the same coloured codes as in the flow chart highlighting the power profiles of the different stages of the devices under analysis. It can be seen that the consumption of the controller in run mode is present all the time. On the one hand, the consumption increases when the controller wakes up. This consumption increases, even more when the FFTs are performed. On the other hand, the consumption decreases when

the IMU is measuring and when the controller is sent to an ultra-low consumption mode. Note that the iMX8 controller cannot be sent to any low power mode, so the power profile is expected to be nearly constant.

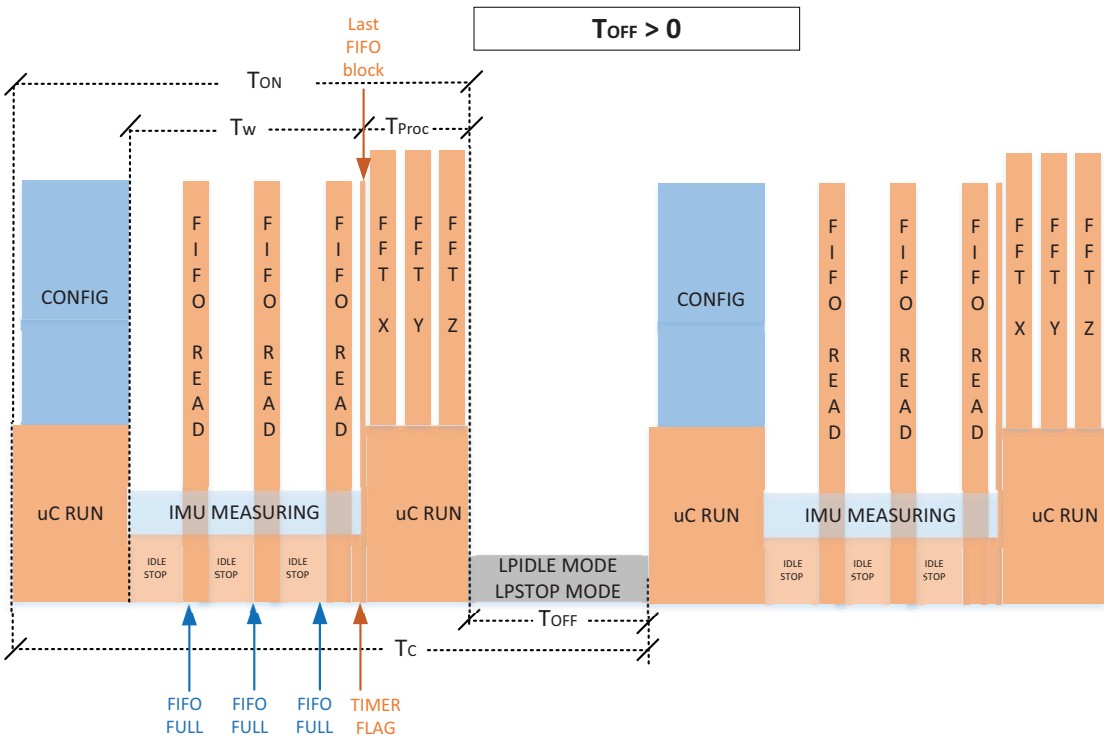

**Figure 9.** Power profile when $T_{OFF} > 0$.

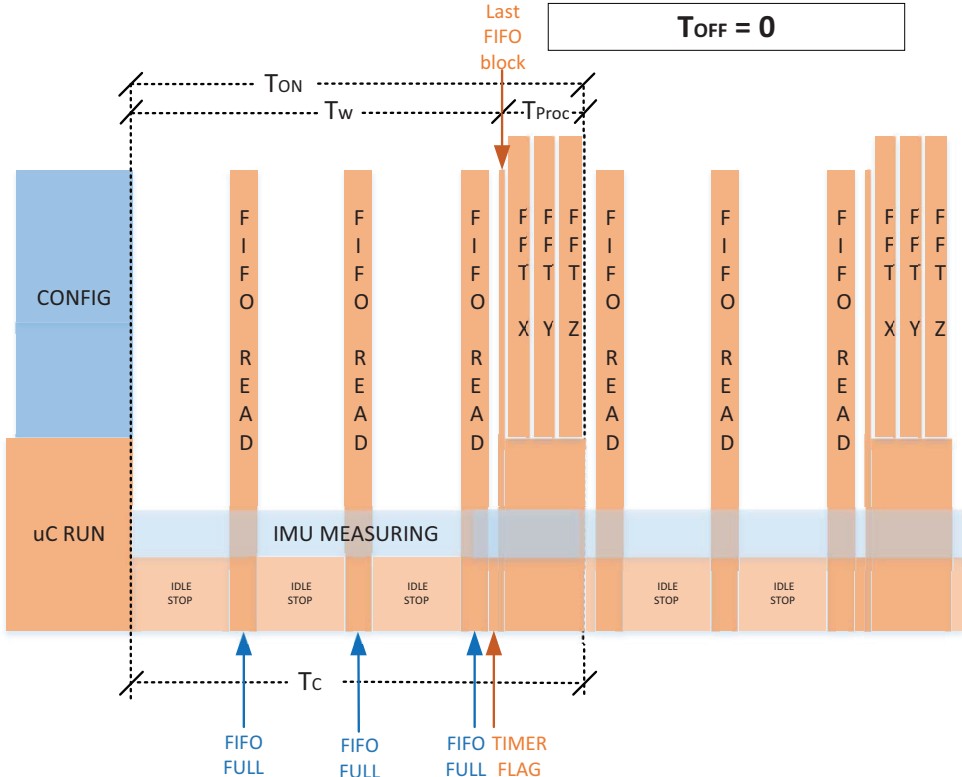

**Figure 10.** Power profile when $T_{OFF} = 0$.

When $T_{OFF} = 0$, then $T_{ON} = T_C$. In this case, as it is shown in Figure 10, the controllers do not need to switch to the sleep mode, and thus, the $T_{ON}$ time is slightly reduced. Moreover, the RT takes approximately 1 ms for the controller to return from the IDLE state. If the FIFO is filled, an interruption is generated, and then the controller should be able to access the first data stored in the FIFO before new data is stepped on. We thus will not send the controller to the IDLE state when the IMU is working at sampling frequencies ($F_s$) equal or greater than 833 Hz.

For the performance measurements, we have to consider that the processing time ($T_{proc}$) is the time it takes to the processor to read the last FIFO block plus the time it takes to process the FFTs, i.e., the time it takes to the processor to process the FFTs from the TIMER FLAG is set.

*5.3. Performance Measurements*

Figure 11 shows the performance of the ST and the RT controllers. It can be seen that there is a significant difference in the performance between ST and RT. This performance is more evident for FFTs with more number points. The higher the number of FFT points (nFFT), the higher the computational cost, and thus, it will be more pronounced in the ST controller than in the RT one. We can see that, in the case of the RT controller, the processing time ($T_{proc}$) slope starts with a 4 ms for an nFFT = 64 points and ends with a 60 ms for an nFFT = 1024 points, approximately. In the case of ST, the slope starts with 32 ms for an nFFT = 64 points and ends with 800 ms for an nFFT = 1024 points, approximately.

The slight differences between the processing times ($T_{proc}$) at different sampling frequencies ($F_s$) are related to the FIFO reading time. From the moment the interruption by the TIMER FLAG is generated until the FIFO is read, a short time elapses in which new data continues to be stored in the FIFO. The higher the ($F_s$), the greater the number of data stored in that short period. Therefore, the longer the time it will take to read the FIFO. As well as processing data, RT is also faster to execute read operations. Figures 12 and 13 show the ST and RT controllers' performance for different sampling frequencies ($F_s$). In addition, it can be seen how long the processing time ($T_{proc}$) remains to the limit determined by the TIMER FLAG or the time it takes to fill the FIFO, i.e., the TIMER FLAG time and FIFO FULL time determine the maximum time permissible to the controller be able to process whole data each cycle by supposing no sleep time.

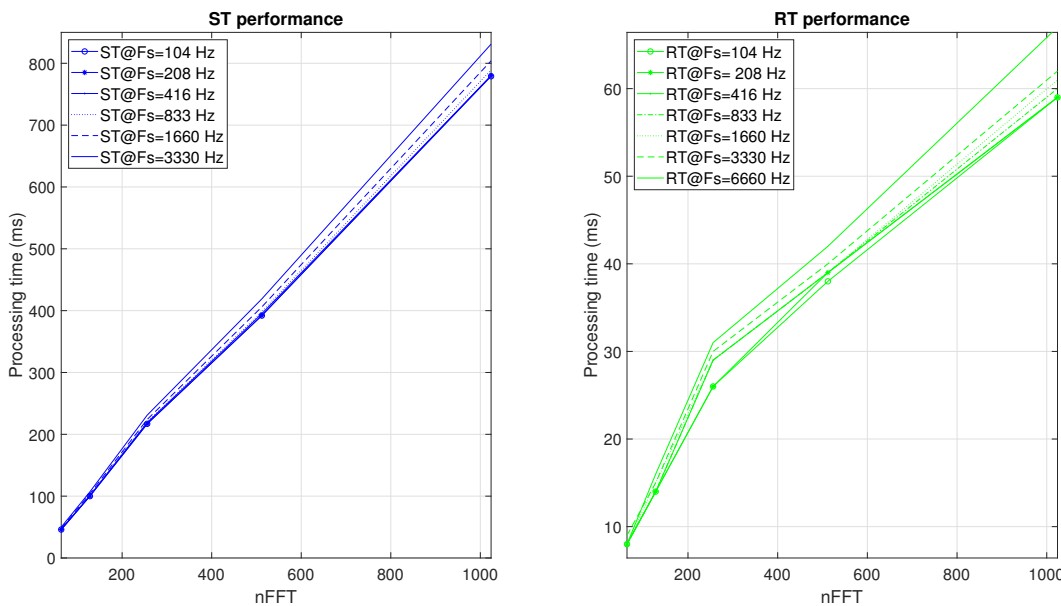

**Figure 11.** Performance measurements.

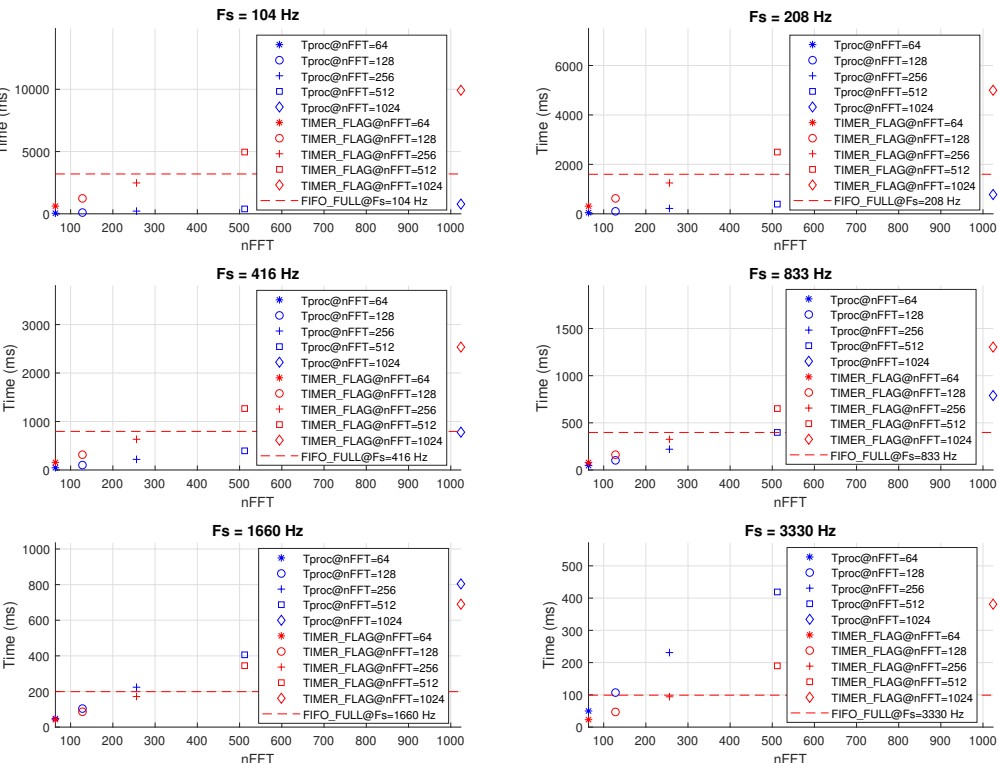

**Figure 12.** ST Performance measurements.

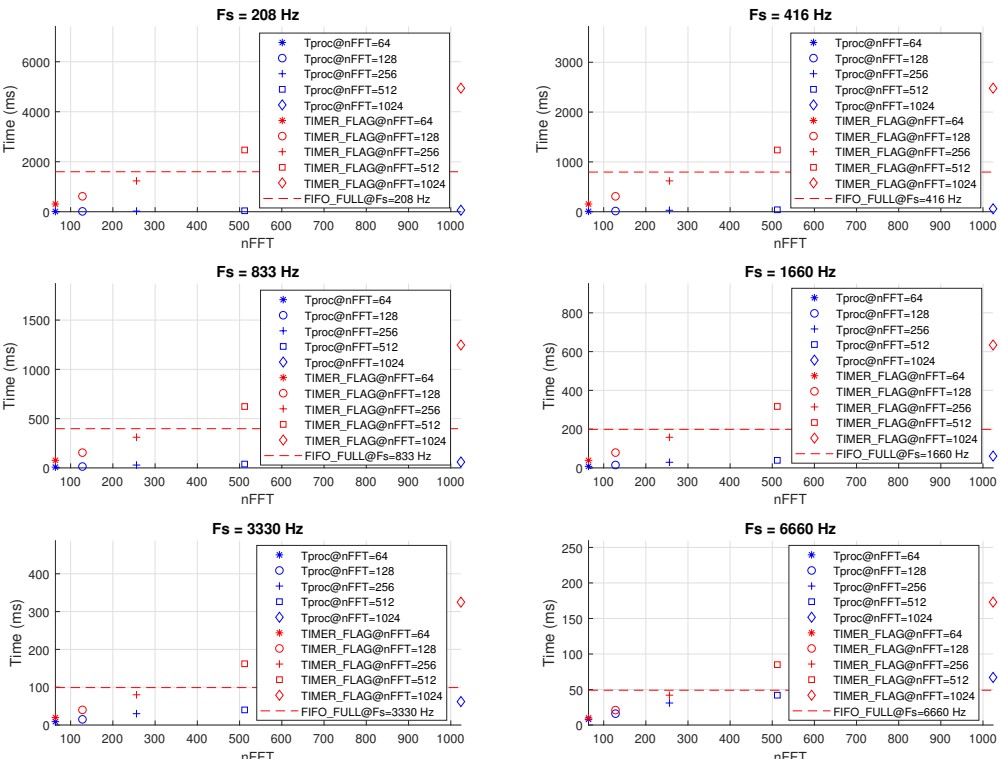

**Figure 13.** RT Performance measurements.

From Figure 12, we can see that for the ST controller, for a $F_s$ = 833 Hz and nFFT = 512, the processing time ($T_{proc}$) is almost at the FIFO FULL time limit. From that point on, in the case that there is no sleep time, either because the processing time ($T_{proc}$) exceeds the limit set by the TIMER FLAG or the FIFO FULL time, the controller would not be able to process the data every cycle. In the case that the controller is set to sleep mode, it could process the

data before a new cycle starts if the $T_{OFF}$ time is long enough. From Figure 13, we can see that for the RT controller, for a $F_s$ = 6660 Hz and nFFT = 1024, the processing time ($T_{proc}$) exceeds the FIFO FULL time limit and thus, only for this case, the controller would not be able to process the data every cycle (in the case that there is not sleep time). Unlike the other controllers, in the case of the iMX8, the control of the IMU is done using high-level libraries. This results in a loss of control by the programmer and makes it challenging to optimise the code for performance improvement. In addition, being running under an OS makes the execution time of this process very random. Table 3 shows a performance comparison between the iMX8 and the RT controllers. It can be seen that the iMX8 processes any of the FFTs faster. However, reading data from the IMU is much slower. The full FIFO read time in the case of the iMX8 is random within this range. This randomness leads to errors in the execution of the flow, which are accentuated as the sampling frequency is increased.

**Table 3.** iMX8 Vs RT performance comparison.

| nFFT | 64 | 128 | 256 | 512 | 1024 |
|---|---|---|---|---|---|
| iMX8 3 FFT processing (ms) | <1 | <1 | 1 | 3 | 9 |
| RT 3 FFT processing (ms) | 4 | 6 | 13 | 27 | 57 |
| iMX8 FIFO FULL reading (ms) | | | 70–120 | | |
| RT 3 FIFO FULL reading (ms) | | | 22 | | |

*5.4. Power Consumption Measurements*

Tables 4 and 5 show the average power consumption of each type of event that is generated on each cycle, both for the ST and RT controllers, respectively.

**Table 4.** Average power consumption of the events generated in the ST controller power profile.

| $\overline{P_{OFF}}$ | CONFIG | STOP + IMU MEAS | FIFO READ | FFT | uC RUN |
|---|---|---|---|---|---|
| 3.25 mW | 347.5 mW | 137 mW | 225 mW | 118.5 mW | 115 mW |

**Table 5.** Average power consumption of the events generated in the RT controller power profile.

| $\overline{P_{OFF}}$ | CONFIG | IDLE + IMU MEAS | FIFO READ | FFT | uC RUN |
|---|---|---|---|---|---|
| 12.2 mW | 55 mW | 109 mW | 281 mW | 329 mW | 259 mW |

In the case of the iMX8, the consumption is practically constant, regardless of the number of FFT points (nFFT), the sampling frequency ($F_s$) or the duty cycle. In addition, the slight differences that may exist are negligible compared to the base consumption of the iMX8 itself. The average consumption of the iMX8 when the program is running is approximately 3.47 W. Therefore, a comparison of the energy consumed by the proposed architecture when ST or RT controller is the main processor has been carried out. To do this, we have considered two operating modes. The first one is when the controller is sent to the sleep mode such that $T_{OFF} > 0$ and the second one is when the controller is continuously processing data such that $T_{OFF} = 0$.

5.4.1. With Sleep Mode ($T_{OFF} > 0$)

For the operating mode $T_{OFF} > 0$, Figure 14 shows the energy consumption during $T_{ON}$ as a function of the number of FFT points (nFFT) and for different sampling frequencies ($F_s$). We can distinguish three cases:

- $F_s$ < 833 Hz: In the RT, FIFO READ and FFT events consume more than ST ones. However, these are faster in the RT. Moreover, IDLE mode consumes less than STOP mode. For these reasons, the energy consumed during $T_{ON}$ by the RT controller is lower than for the ST one.

- $F_s$ = 833 Hz: The energy consumed during $T_{ON}$ by the RT controller is more significant than for the ST one because it is not possible to send the RT to IDLE mode.
- $F_s$ > 833 Hz: Despite not being able to send the RT to IDLE mode, $T_{ON}$ decreases with increasing $F_s$, and as the RT is faster, the energy consumed by the RT is also lower than that for the ST one.

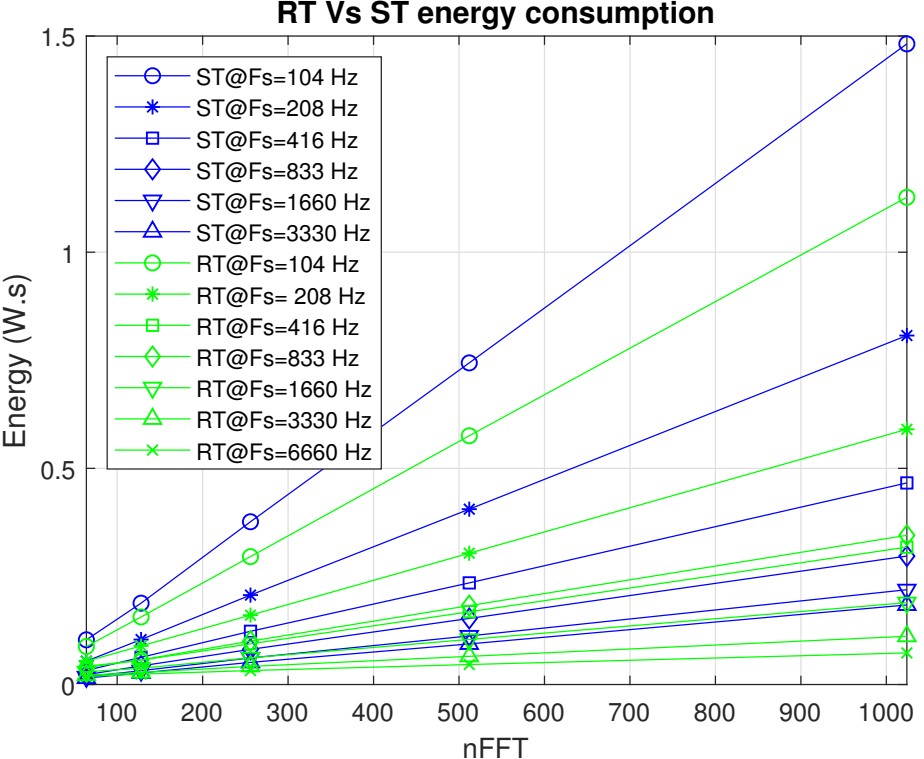

**Figure 14.** Energy consumption measurements during $T_{ON}$.

Considering these results and the FR8RAIL use case, we aim to estimate the autonomy of the proposed architecture with a battery of 3.7 V and 10,000 mAh (37 Wh). Figures 15 and 16 show the battery duration for the ST and RT controllers as a function of $T_{OFF}$ and for different sampling frequencies ($F_s$). It can be seen that the off-time $T_{OFF}$ is the parameter that mostly determines the battery life, that is, the longer the $T_{OFF}$, the longer the time spent in ultra low power mode and, therefore, the lower the average energy consumption per cycle. In this way, the longer the $T_{OFF}$, the greater the difference between the battery life for the ST than the RT one. This is logical since consumption in $T_{OFF}$ is lower in ST than in RT. However, when $T_{OFF}$ tends to zero, the battery life tends to be larger for the RT than for the ST one. On the other hand, battery life increases as the sampling frequency ($F_s$) decreases. This is also logical since the system remains powered on for a longer time as it has to capture the same data at a lower rate.

For the FR8RAIL use case, where a 5 min cycle time is required, with a $F_s$ = 3300 Hz and an FFT of 1024 points, the estimated battery life is 122 days for the RT and 400 days for the ST.

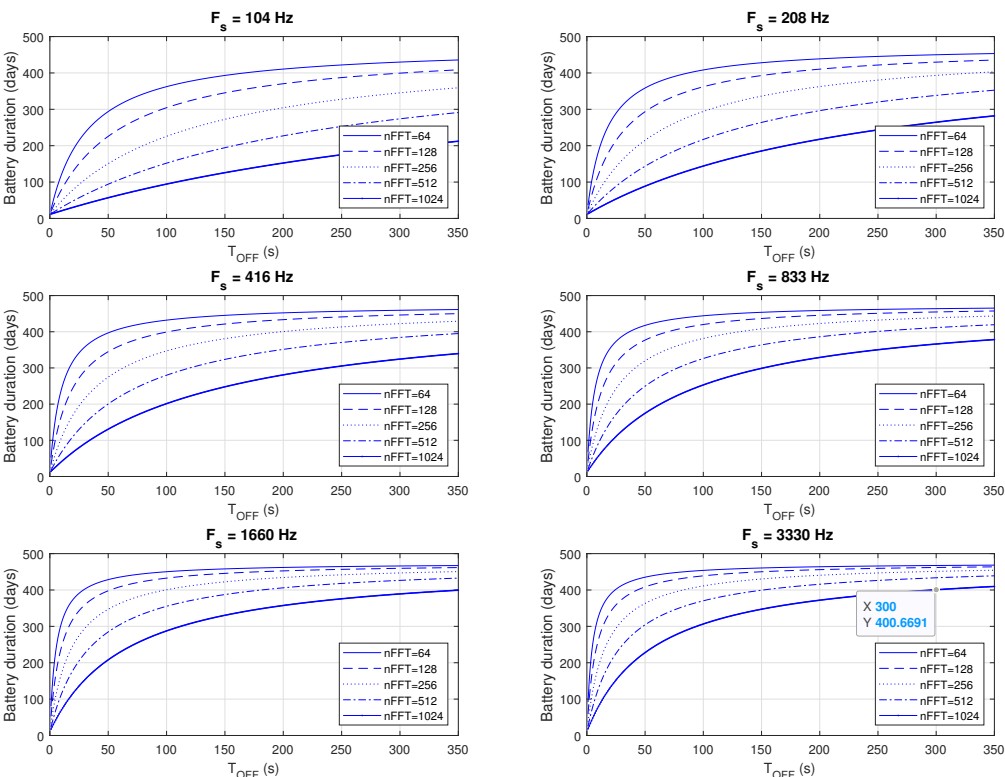

**Figure 15.** Battery duration for the ST controller and operating mode $T_{OFF} > 0$.

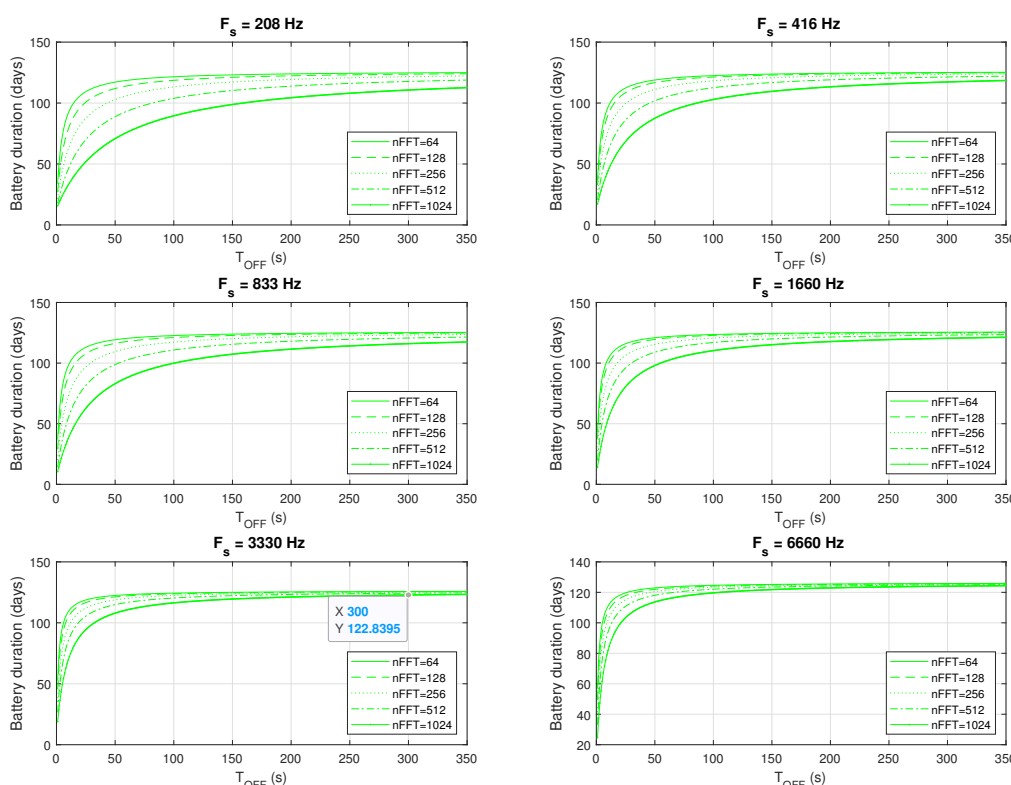

**Figure 16.** Battery duration for RT controller and operating mode $T_{OFF} > 0$.

### 5.4.2. Without Sleep Mode ($T_{OFF} = 0$)

For the operating mode $T_{OFF} = 0$, $T_{ON} = T_C$, and thus, it makes less sense to calculate energy per cycle. Figure 17 shows the power consumption during each cycle $T_C$ as a

function of the number of FFT points (nFFT) and for different sampling frequencies ($F_s$). In the case of the ST controller, it is only shown the power consumption for sampling frequencies ($F_s$) lower than 416 Hz because it cannot perform the processing at higher frequencies at each cycle. It can be seen that the RT consumes lower power than ST for the same sampling frequencies ($F_s$). Since sending the RT to the IDLE mode is impossible, the power consumption increases for sampling frequencies ($F_s$) greater than 833 Hz.

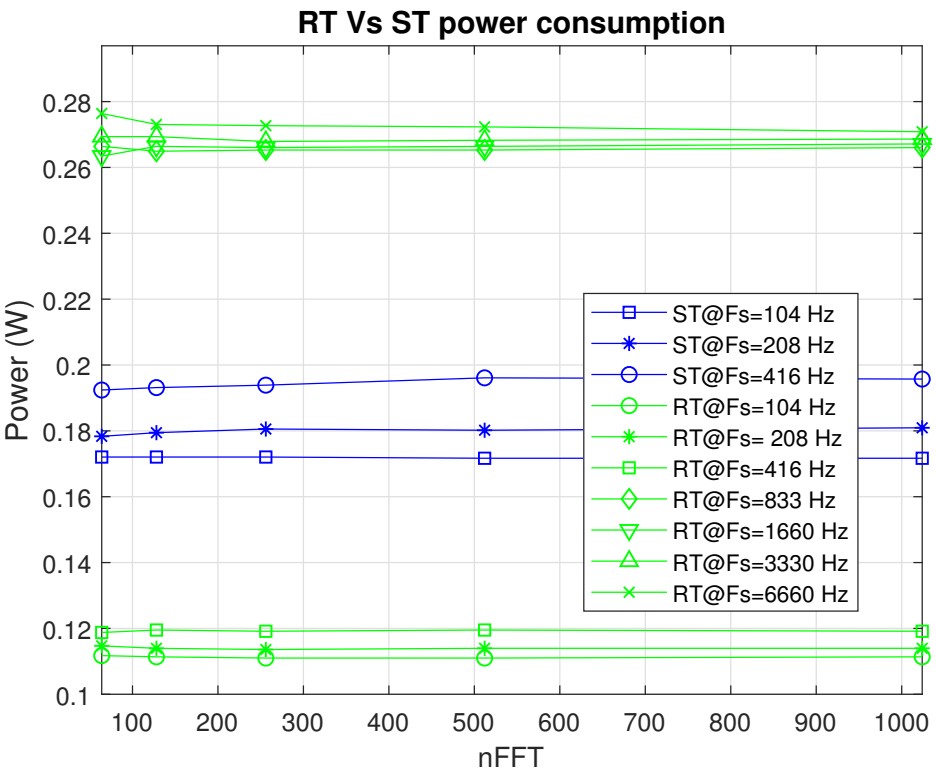

**Figure 17.** Power consumption measurements for the operating mode $T_{OFF} = 0$.

Considering such power consumption and the FR8RAIL use case, the battery life has been estimated for different sampling frequencies $F_s$ and a number of FFT points (nFFT). Tables 6 and 7 show the battery duration for the ST and RT controllers, respectively. Note that this mode of operation requires continuous processing and is, therefore, high energy demanding, which should be nuanced for the representative functions presented in the first sections. Better management of the internal processing and energy harvesting techniques could be considered. Still, it is possible to achieve a battery life, in the worst case, of up to several days, taking into account that the battery mounted is a small 10,000 mAh battery. It can be seen that for the FR8RAIL use case if the system were kept in this mode of operation, with a $F_s$ = 3300 Hz and an FFT of 1024 points, a battery life of 137 h will be expected for the RT controller.

**Table 6.** Battery duration in hours for ST controller and operating mode $T_{OFF} = 0$.

|  | $F_s$ = 104 Hz | $F_s$ = 208 Hz | $F_s$ = 416 Hz |
|---|---|---|---|
| nFFT = 64 | 215 | 207 | 192 |
| nFFT = 128 | 215 | 206 | 191 |
| nFFT = 256 | 215 | 205 | 190 |
| nFFT = 512 | 215 | 205 | 189 |
| nFFT = 1024 | 215 | 204 | 189 |

**Table 7.** Battery duration in hours for RT controller and operating mode $T_{OFF} = 0$.

|  | $F_s$ = 104 Hz | $F_s$ = 208 Hz | $F_s$ = 416 Hz | $F_s$ = 833 Hz | $F_s$ = 1660 Hz | $F_s$ = 3330 Hz | $F_s$ = 6660 Hz |
|---|---|---|---|---|---|---|---|
| **nFFT = 64** | 333 | 324 | 310 | 139 | 139 | 137 | 135 |
| **nFFT = 128** | 333 | 324 | 310 | 139 | 139 | 137 | 135 |
| **nFFT = 256** | 333 | 324 | 310 | 139 | 139 | 137 | 135 |
| **nFFT = 512** | 333 | 324 | 310 | 139 | 139 | 137 | 135 |
| **nFFT = 1024** | 333 | 324 | 310 | 139 | 139 | 137 | 135 |

## 6. Coverage Results

To assess the communication technology of this connected heterogeneous platform, a test campaign was organised in Sweden, hosted by Trafikverket and coordinated by SNCF. The objective of the test campaign was to test LoRa technology to set the ground for discussion and possible agreement on a common communication standard for IoT in rail freight. Only the LoRa radio layer is implemented in the design, and no protocol layer is implemented. As only mono-frame messages are transmitted, and no retransmission protocol is implemented, the Packet Error Rate (PER) calculation will be directly the ratio of messages lost.

The tests were performed during four days in Sweden and allowed to test LoRa solution in a real environment. Figures 18 and 19 show the distribution of the nodes over the train. The test was divided into two phases:

- Static tests to:
  - Check installation.
  - Evaluate range of communication on a real train.
- Dynamic test to:
  - Expose the communication systems in real railway environment.
  - Generate enough messages to build statistical analysis based on logs recorded.

For all the tests (static and dynamic), each node was used in only one direction of communication:

- Master node on wagon 1: only reception of messages coming from the 5 nodes deployed on train.
- Node 1 to node 5: only transmission of messages to master node on wagon 1.

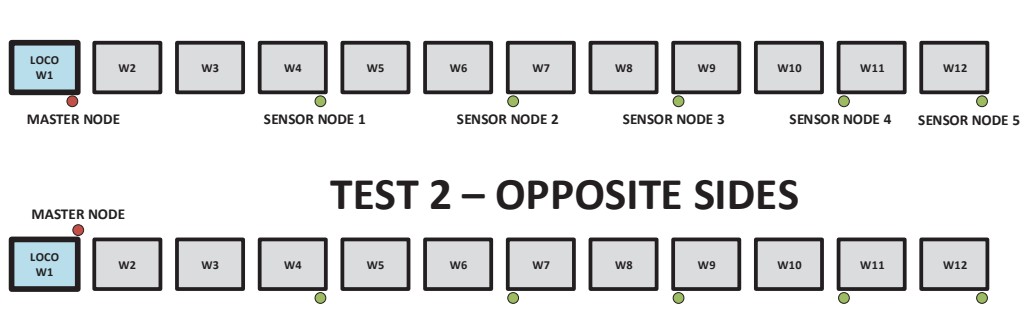

**Figure 18.** Distribution of the nodes over the train for LoRa tests.

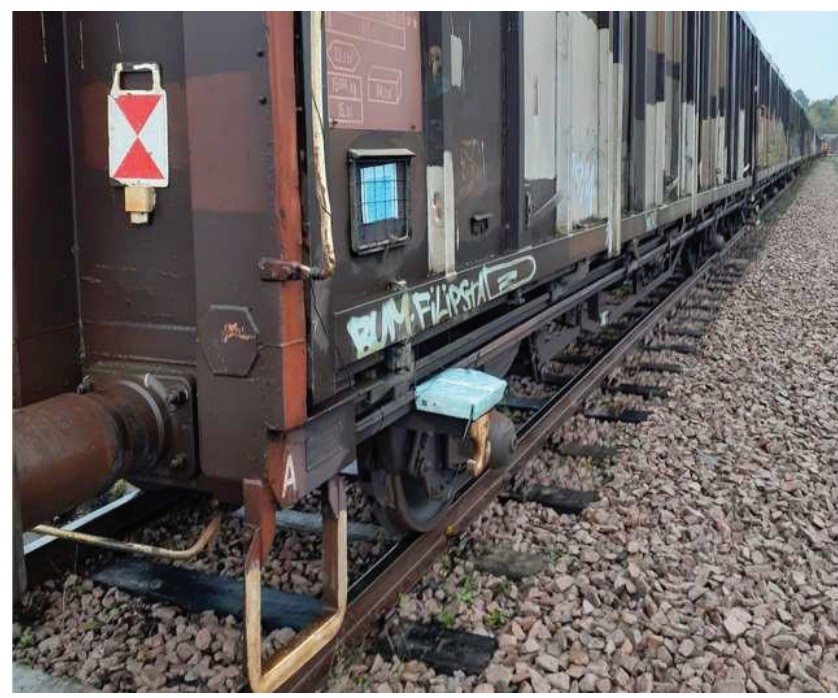

**Figure 19.** Placement of the nodes in the wagon.

Main observation: Table 8 shows the results for the static test 1. LoRa Radio technology aims to cover 12 wagons with an acceptable loss ratio when installed on the same side. SNR decreases when the distance is higher; this train length (11 wagons to jump) does not impact the loss ratio.

**Table 8.** LoRa Radio Communication Static Test 1. Communication in line of sight.

|  | **Sensor Node 1** | **Sensor Node 2** | **Sensor Node 3** | **Sensor Node 4** | **Sensor Node 5** |
|---|---|---|---|---|---|
| **Packets sent** | 57 | 57 | 57 | 57 | 57 |
| **Packets received on Master** | 57 | 55 | 56 | 55 | 55 |
| **Packets not received** | 0 | 2 | 1 | 2 | 2 |
| **Packet loss ratio** | 0 | 3.51 | 1.75 | 3.51 | 3.51 |

Main observation: Table 9 shows the results for the static test 2. The results for communication on opposite sides are better than previous results when communicating in line of sight. Some loss starts to be observed when transmission jumps over nine wagons or more with an acceptable loss ratio of 2%.

**Table 9.** LoRa Radio Communication Static Test 2. Communication in Opposite sides.

|  | **Sensor Node 1** | **Sensor Node 2** | **Sensor Node 3** | **Sensor Node 4** | **Sensor Node 5** |
|---|---|---|---|---|---|
| **Packets sent** | 49 | 49 | 49 | 49 | 49 |
| **Packets received on Master** | 49 | 49 | 49 | 48 | 48 |
| **Packets not received** | 0 | 0 | 0 | 1 | 1 |
| **Packet loss ratio** | 0 | 0 | 0 | 2.04 | 2.04 |

The dynamic test period was from 6 October 2021 at 06:37:44 UTC to 7 October 2021 at 07:33:00 UTC. The train was in motion 282 min out of 1496 min of the test (19% of time).

During this time, wagons travelled 272 km. The environmental conditions met during the test were:

- Forest;
- Cities;
- Marshalling yards;
- Curves;
- Lake proximity.

Dynamic test analysis shows comparable results (Table 10) to those observed during the static test. Focusing on transmissions from wagon 12 to wagon 1:

- Loss ratio observed is 3.03% instead of 2.04% during static test.
- No degraded performances due to potential interferences were observed.
- No important increasing of loss ratio due to railway environment.

**Table 10.** LoRa Radio Communication Dynamic Test.

|  | Sensor Node 1 | Sensor Node 2 | Sensor Node 3 | Sensor Node 4 | Sensor Node 5 |
|---|---|---|---|---|---|
| **Packets sent** | 1484 | 1484 | 1484 | 1484 | 1484 |
| **Packets received on Master** | 1449 | 1470 | 1465 | 1384 | 1440 |
| **Packets not received** | 36 | 15 | 20 | 101 | 45 |
| **Packet loss ratio** | 2.42 | 1.01 | 1.35 | 6.8 | 3.03 |

Tests in Sweden allows confirming that LoRa is working in a real railway environment with good results regarding coverage (12 wagons in a single hop) and mean loss ratio of between 2% and 3%.

## 7. Discussion and Conclusions

The digitisation of freight rail is an essential improvement to create modern functions that offer a cost-effective, attractive service and improved operational opportunities to operators. These modern functions need intelligence, detection, actuation and communications, requiring power supply, which is not granted in a freight wagon. This is the reason why the assessment of the communication technologies and the power consumption depending on the computing device is crucial. For that goal, a connected heterogeneous multiprocessing wireless architecture has been proposed, implemented, tested and stressed. Thanks to the heterogeneous nature of this architecture and its configurability, it allows us to select the most suitable computing device and data analysis and communication strategy in terms of efficiency and performance of the functions that it needs to host and support. With this approach, operation data are reported to the centralised Freight Driver Assistant System with a diverse computational cost, linking it to the energy consumption and the robustness and coverage. It is possible to process raw data in the Edge and send meaningful data over a low power consumption communication technology such as LoRa or send the raw data directly if they fit into the communication channel and the duty cycle of the operative function.

This paper proposes a configurable HW and FW architecture with, among other technologies, three types of controllers, which cover a wide range of possibilities in terms of processing capacity and energy consumption. For the high-level data communication FR8RAIL use case, this HW has been implemented in a star wireless network topology. There is a coordinator in the locomotives, named LOBU, and distributed nodes in the wagons, called Wagon On Board Units. The WOBU uses the low power ST microcontroller as the coordinator for the configuration of the resources needed from the platform. For the less demanding functions, such as the cargo monitoring system, the implementation uses only the ST microcontroller as the processor. The low power consumption of the platform with that configuration the proposed operating mode are decisive to ensure that the battery life is prolonged over the years. In the measurements carried out in Sweden, the average

power consumption of the node has been 3.16 mW, sending data through the LoRA link every 5 min. With these results, battery life of 1.9 years is estimated. Moreover, coverage results have confirmed that LoRa is working in a railway environment with good results regarding coverage (12 wagons in a single hop) and mean loss ratio of between 2% and 3%.

In order to be able to support other more processing demanding functions already identified for the digitalisation of the rail freight transport, it is necessary to analyse the performance and power consumption of the different controllers, as proposed by this paper. The authors of this paper have considered the Fast Fourier Transform of the acceleration data for the X, Y and Z axes as a representative process for the Wagon Monitoring System. It has been implemented in the three types of controllers. For these three controllers, different tests have been carried out by varying the parameters of the implementation, such as the duty cycle of the function, through the off-time ($T_{OFF}$), the sampling frequency of the data acquisition ($F_s$) and the number of FFT points. However, the iMX8 is faster than the other controllers for more data and points. Other factors such as the OS or the interaction with the peripherals produce uncertainties, making it impossible to follow the correct flow of the program deterministically. For intensive processing during a short period, this controller would be interesting. However, for the processing in which strict time control is necessary, it would not be recommended to use it or use it under the generic OS mounted. In addition, the system's energy consumption shoots up to 3.47 W, which would discharge the 37 Wh battery in a matter of 10 h approximately. These reasons have led us to present the comparative analysis between ST and RT controllers.

The RT controller is clearly superior to the ST in terms of performance. This becomes more apparent as the number of FFT points increases. The RT controller executes an FFT of 1024 points in 57 ms while the ST controller does in 770 ms. Furthermore, for the mode of operation in which the controller is not put into the sleep mode, the ST controller would not be able to process the data every cycle for sampling frequencies higher than 833 Hz. In terms of power consumption, if the controller is not set to the sleep mode, the RT controller is less power demanding than the ST controller for any parameter configuration. For a sampling frequency less than 833 Hz, the node battery life is expected to be around eight days when the ST is processing and about 12 days when the RT is processing, both at continuous acquisition and processing. That would be the case for a Train Integrity function, for instance.

In the operating mode where the controller is set to the sleep mode, the off-time (TOFF) is the parameter that mainly determines the battery life. In almost all cases, the battery duration is higher for the ST controller. This is logical since consumption in TOFF is lower in ST than in RT. For a FR8RAIL function such as Wagon Monitoring system, by supposing a cycle time ($T_C$) of 5 min, for an FFT of 1024 points and a sampling frequency of 3330 Hz, the node battery life is expected to be around 400 days when the ST is processing and about 122 days when the RT is processing.

Given the obtained results, combining different processors into the connected heterogeneous multiprocessing platform allows us to define optimal performance and consumption depending on the functions to be developed and deployed on the wagons and trains. It is possible to drive different tasks or processes to any of these processors to find the optimum balance between performance and consumption. The connected heterogeneous multiprocessing platform beneficially distributes the functions among the processors and determines cases where Edge Computing or raw data reporting are optimal for the WOBU units.

## 8. Patents

As a result of this Shift2Rail IP5 FR8RAIL projects, the authors of this paper are using the elements shown in this paper to propose a patent dedicated to the Automatic Brake Test operation, which is one of the key functions for the freight train preparation. Further steps need to be done prior to that protection, though.

**Author Contributions:** Conceptualization, M.L.; methodology, M.L., I.A., A.P. and R.C.R.; software, M.L. and A.P.; validation, M.L., R.C.R. and A.P.; formal analysis, M.L., A.P. and I.A.; investigation, M.L., I.A., A.P. and R.C.R.; writing—original draft preparation, M.L., I.A., A.P. and R.C.R.; writing—review and editing, all authors; project administration, J.M. and I.A.; funding acquisition, J.M. All authors have read and agreed to the published version of the manuscript.

**Funding:** This research was funded by FR8RAIL-3, which has received funding from the European Union's Horizon 2020 research and innovation programme under grant agreement No: 881778. This research also received funding by FR8RAIL-4, which has received funding from the European Union's Horizon 2020 research and innovation programme under grant agreement No: 101004051.

**Conflicts of Interest:** The authors declare no conflict of interest.

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
