# Peer review of "Connected Heterogenous Multi-Processing Architecture for Digitalization of Freight Railway Transport Applications"

_electronics, doi:10.3390/electronics11060943_

Round 1
Reviewer 1 Report
The authors focused on two main problems of the Freight Wagon diagnostic which is a calculation of the FFT and LoRa communication. The solution of the first problem is solved in more detail than the second one.
The proposed hardware is rather complicated. The proposed hardware consists of different sensors and microcontrollers, that are controlled by a low-power microcontroller and that exchange the data via the standard busses. Just the data exchange is slow and complicated.
There are the heterogenous multicore microcontrollers that are designed for this task in which the low power consumption of the system is required and the system also disposes of high computation power for the massive computing. In addition, the communication between the microcontroller cors is very fast.
The next comment concerns the FFT implementation. It is not clear whether FFT is calculated in fixed or floating-point arithmetic. The discussion concerning energy or power efficiency is a little foggy. The authors should calculate the energy consumption and the calculation time of each microcontroller.
The next possibility of implementation of the FFT is FPGA. There are specialized highly optimized cores for this. The comparison of the implementation to the microcontroller and FPGA would dramatically increase the scientific soundness of the paper.
The last comment is on the LORA communication. LORA operates in ISM radio band. The access to this band is very weakly regulated. The interference from the other user cannot be avoided. It is on the system integrator whether he accepts this level of service. There are other possibilities based on dedicated UHF or VHF radio channels. The authors should consider this alternative to the paper.
Author Response
The authors focused on two main problems of the Freight Wagon diagnostic which is a calculation of the FFT and LoRa communication. The solution of the first problem is solved in more detail than the second one.
In our view, the calculation of the FFT or the communication by LoRa are not the main challenges presented by the modernization and the diagnosis of the freight railways. Both are tools that can be used by the proposed HW architecture. One of the purposes of this article is to show a heterogeneous multiprocessor architecture in which it is possible to implement applications that could be required by the digitization of a freight train. The calculation of the FFT has been proposed as an example, as it is a complex and very common mathematical operation in many algorithms. Also, as stated several times in the article, the goal of using the FFT is not to evaluate its own implementation itself, but rather the goal is to observe the consumption and performance of the different controllers under a complex mathematical operation of this type.
The proposed hardware is rather complicated. The proposed hardware consists of different sensors and microcontrollers, that are controlled by a low-power microcontroller and that exchange the data via the standard busses. Just the data exchange is slow and complicated.
There are the heterogenous multicore microcontrollers that are designed for this task in which the low power consumption of the system is required and the system also disposes of high computation power for the massive computing. In addition, the communication between the microcontroller cors is very fast.
Certainly, you are completely right. There are multicore processors like the one in the iMX8, which has a 4x Cortex-A53 core platforms up to 1.8GHz per core and a 1x Cortex-M4 core up to 400MHz. However, although the Cortex A53 is capable of performing very high performance operations, it has problems executing tasks in real time, even more when it works under an operating system. This reason leads the manufacturer NXP to include the Cortex M4 to carry out tasks of this type. Therefore, with an architecture like the iMX8 we could not cover a wide range of real time applications. With the proposed architecture shown in the paper, it is possible to implement applications that require real-time execution with low consumption and low computational cost, applications with higher computational cost or even operate under an operating system. A wide range of possibilities are covered in that way.
The next comment concerns the FFT implementation. It is not clear whether FFT is calculated in fixed or floating-point arithmetic.
As indicated in the above answer, the calculation of the FFT has been proposed as an example. No information is given on how it is calculated because one of the objectives of the article is to evaluate the performance and consumption of the different controllers for the same mathematical operation. In any case, the operation is calculated in floating point. So, since the ST microcontroller does not have an FPU (Floating Point Unit), unlike the IMXRT or iMX8, the calculation is done by the compiler itself by means of code, and thus the performance is reduced.
The authors should calculate the energy consumption and the calculation time of each microcontroller.
It is our understanding that this is what has been really done. The time is presented for each operation that each controller performs as well as the power consumed in each of these operations.
The next possibility of implementation of the FFT is FPGA. There are specialized highly optimized cores for this. The comparison of the implementation to the microcontroller and FPGA would dramatically increase the scientific soundness of the paper.
Thanks for the comment. If the application consisted of making FFTs, an FPGA would have been implemented in the HW to carry out this task. You are absolutely right on that point. In that case, this would have made the system rigid. In the presented proposal, the HW is flexible due to its heterogeneous character and allows a wide range of applications useful for the transition stpes that are faced these days on the freight railway sector.
The last comment is on the LORA communication. LORA operates in ISM radio band. The access to this band is very weakly regulated. The interference from the other user cannot be avoided. It is on the system integrator whether he accepts this level of service. There are other possibilities based on dedicated UHF or VHF radio channels. The authors should consider this alternative to the paper.
For the use case presented, it is true that there are more suitable technologies for communications. The decision to use LoRa was due to the fact that it allowed us two modes of use, one of them LoRa Radio P2P, to communicate data from the wagons to the loco when the train in on the mainline (tracks). And the other, using the LoRaWAN protocol to be able to transmit data to a commercial gateway located in the Train Stations/depots when the train has arrived to them.
The evaluation of this solution was highly recommended by the train operator. And been the ISM band one of the weakest points, the results obtained during the tests reports in the paper show us that LoRa behaved robustly in terms of packet loss and interference.
Reviewer 2 Report
The paper designs a wireless connected heterogeneous multiprocessing architecture that ensures suitable computing resources are provided for improved efficiency and overall network performance.
The paper is interesting in terms of high-level freight railway transportation applications. The reviewer has the following comments to be addressed by the authors:
- The abstract is supposed to briefly highlight the main contributions of this research. However, the reviewer is unable to grasp such information from the abstract in its current form. For example; what does the paper seek to propose? How is the proposed architecture used to ensure the objectives of the research? What approaches are used as baselines for performance comparison? The authors should provide such information in the abstract.
- In Section 1, the authors discuss a number of related works. However, the limitations of these existing works are not mentioned. What are the shortfalls of the existing literature and how does this current research mitigate such shortfalls?
- On page 14, the authors state “….and when it is not send (T_OFF=0)”. What does not sending the architecture to sleep mode mean?
- In subsection 4.3, the authors explain the trend shown in Figure 11. However, the values and points used for the explanation are untraceable. It would be better if the authors made references to values that can be traced on the figure.
- The organization of the manuscript can be improved. The paper is very difficult to follow. Also, there are a lot of grammatical and editorial errors in the current manuscript. The authors should carefully proofread the whole manuscript to correct all errors.
Author Response
The abstract is supposed to briefly highlight the main contributions of this research. However, the reviewer is unable to grasp such information from the abstract in its current form. For example; what does the paper seek to propose? How is the proposed architecture used to ensure the objectives of the research? What approaches are used as baselines for performance comparison? The authors should provide such information in the abstract.
Thanks for the comment. The abstract has been improved within the 200 words imposed by the format of the journal. It emphasizes now the main points presented in the paper even if it is quite constrained by the space offered. More details could be found in the conclusions and discussions
In Section 1, the authors discuss a number of related works. However, the limitations of these existing works are not mentioned. What are the shortfalls of the existing literature and how does this current research mitigate such shortfalls?
After your comment, we find it appropriate to rewrite the section to more clearly establish the importance of the related works and the purpose of our research work. We begin by clearly defining the main problems to be solved in freight railway monitoring systems and the technologies that could be implemented to solve them. Then, several examples of IoT and edge computing application projects in freight railways are presented. Finally, the reconfigurable hardware edge computing projects closest to our proposal are mentioned. It is established that the convenience of reconfigurable hardware, based on heterogeneous multiprocessing architecture with microcontrollers has not been explored yet.
We hope that this could also contribute to a better understanding of the organization of the overall paper, as answered to your last comment
On page 14, the authors state “….and when it is not send (T_OFF=0)”. What does not sending the architecture to sleep mode mean?
The sentence in page 14 is understandable with the explanations in page 11
T_OFF is the time in which the node remains in sleep mode.
The T_OFF is one of the input parameters for the proposed system, as it is seen in figure 7. The fact of setting T_OFF >0 implies that the system will be in sleep mode during the time determined by T_OFF until a new iteration of measurement and processing occurs. This gives the controller more time to perform the FFT processing before starting a new measurement. Setting T_OFF = 0 implies not sending the system to sleep mode, which means that a sensor such as the IMU would be continuously measuring and processing new FFTs.
A clearer explanation has been proposed just above Figure 8 to improve the understanding of that concept.
In subsection 4.3, the authors explain the trend shown in Figure 11. However, the values and points used for the explanation are untraceable. It would be better if the authors made references to values that can be traced on the figure.
Thank you very much for that comment. A new figure has been created for the sake of understanding, with clearer differences among the curves.
The organization of the manuscript can be improved. The paper is very difficult to follow. Also, there are a lot of grammatical and editorial errors in the current manuscript. The authors should carefully proofread the whole manuscript to correct all errors.
The organization of the paper is now more clearly presented in the last paragraph of the introduction section.
Reviewer 3 Report
The presented manuscript is outstanding. It presents a feasible digitalisation solution for railway transport. The work structure is in compliance with scientific methodology and presents the results in a logical manner. Referenced work is also suitable with the paper's topic. Moreover, the presented work offers sufficient details to replicate the results.
Only observation that this reviewer has is regarding the format of the paper. On some pages there is too much space left blank perhaps the authors can fix this (e.g. p.6, 11, 13, 15 to 17, especially p. 18, 22 and 23)
A recommendation for the authors: You stated that it is planned to submit a patent for the presented research. Please make sure that publishing some result before submitting a patent application won't affect the patentability of this research.
Author Response
The presented manuscript is outstanding. It presents a feasible digitalisation solution for railway transport. The work structure is in compliance with scientific methodology and presents the results in a logical manner. Referenced work is also suitable with the paper's topic. Moreover, the presented work offers sufficient details to replicate the results.
Only observation that this reviewer has is regarding the format of the paper. On some pages there is too much space left blank perhaps the authors can fix this (e.g. p.6, 11, 13, 15 to 17, especially p. 18, 22 and 23)
Thanks for that comment. Several references to figures have been updated in the LaTex based file to avoid the spaces created.
A recommendation for the authors: You stated that it is planned to submit a patent for the presented research. Please make sure that publishing some result before submitting a patent application won't affect the patentability of this research.
Also here, thanks for the comment. There has been a though on that and no critical data is revealed in the paper.